# Are Thalamic Intrinsic Lesions Operable? No-Man’s Land Revisited by the Analysis of a Large Retrospective, Mono-Institutional, Cohort

**DOI:** 10.3390/cancers15020361

**Published:** 2023-01-05

**Authors:** Paolo Ferroli, Francesco Restelli, Giacomo Bertolini, Emanuele Monti, Jacopo Falco, Giulio Bonomo, Irene Tramacere, Bianca Pollo, Chiara Calatozzolo, Monica Patanè, Silvia Schiavolin, Morgan Broggi, Francesco Acerbi, Alessandra Erbetta, Silvia Esposito, Elio Mazzapicchi, Emanuele La Corte, Ignazio Gaspare Vetrano, Giovanni Broggi, Marco Schiariti

**Affiliations:** 1Department of Neurosurgery, Fondazione IRCCS Istituto Neurologico Carlo Besta, 20133 Milan, Italy; 2Department of Neurological Surgery, Policlinico “G. Rodolico–S. Marco”, University Hospital, 95123 Catania, Italy; 3Department of Research and Clinical Development, Scientific Directorate, Fondazione IRCCS Istituto Neurologico Carlo Besta, 20133 Milan, Italy; 4Unit of Neuropathology, Fondazione IRCCS Istituto Neurologico Carlo Besta, 20133 Milan, Italy; 5Public Health and Disability Unit–Scientific Directorate, Fondazione IRCCS Istituto Neurologico Carlo Besta, 20133 Milan, Italy; 6Unit of Neuroradiology, Fondazione IRCCS Istituto Neurologico Carlo Besta, 20133 Milan, Italy; 7Department of Paediatric Neuroscience, Fondazione IRCCS Istituto Neurologico Carlo Besta, 20133 Milan, Italy; 8Department of Neurosurgery, Alma Mater Studiorum, University of Bologna, 40126 Bologna, Italy; 9Department of Biomedical Sciences for Health, University of Milan, 20133 Milan, Italy; 10IEN Foundation, 20100 Milan, Italy

**Keywords:** brain tumors, outcome, thalamic gliomas

## Abstract

**Simple Summary:**

Thalamic gliomas are rare neoplasms that represent a major surgical challenge and are characterized by poor postoperative survival. Surgical resection, although associated with improved overall survival (OS), is not always feasible. The aim of our retrospective study was to analyze the associations between possible prognostic factors such as tumor volume, histological grade, the extent of resection, performance status and OS. Surgical removal was demonstrated to be an important prognostic factor when gross total resection/subtotal resection was obtained. Furthermore, patients with a stable 3-month performance status after surgery demonstrated to have a better prognosis in terms of OS. In conclusion, in such kinds of tumors, a precise evaluation of the predictors of the 3-month postoperative Performance Status appears to be crucial in choosing between performing a biopsy or attempting the surgical removal of the tumor.

**Abstract:**

Thalamic gliomas represent a heterogeneous subset of deep-seated lesions for which surgical removal is advocated, although clear prognostic factors linked to advantages in performance status or overall survival are still lacking. We reviewed our Institutional Cancer Registry, identifying patients who underwent surgery for thalamic gliomas between 2006 and 2020. Associations between possible prognostic factors such as tumor volume, grade, the extent of resection and performance status (PS), and overall survival (OS) were evaluated using univariate and multivariate survival analyses. We found 56 patients: 31 underwent surgery, and 25 underwent biopsy. Compared to biopsy, surgery resulted positively associated with an increase in the OS (hazard ratio, HR, at multivariate analysis 0.30, 95% confidence interval, CI, 0.12–0.75). Considering the extent of resection (EOR), obtaining GTR/STR appeared to offer an OS advantage in high-grade gliomas (HGG) patients submitted to surgical resection if compared to biopsy, although we did not find statistical significance at multivariate analysis (HR 0.53, 95% CI 0.17–1.59). Patients with a stable 3-month KPS after surgery demonstrated to have a better prognosis in terms of OS if compared to biopsy (multivariate HR 0.17, 95% CI, 0.05–0.59). Age and histological grades were found to be prognostic factors for this condition (*p* = 0.04 and *p* = 0.004, respectively, chi-square test). Considering the entire cohort, p53 positivity (univariate HR 2.21, 95% CI 1.01–4.82) and ATRX positivity (univariate HR 2.69, 95% CI 0.92–7.83) resulted associated with a worse prognosis in terms of OS. In this work, we demonstrated that surgery aimed at tumor resection might offer a stronger survival advantage when a stable 3-month KPS after surgery is achieved.

## 1. Introduction

The thalamus is a deep-seated and highly eloquent region of the brain that represents a fundamental relay station for the sensory-motor system, although its involvement in higher cortical functions such as memory and language has been widely suggested [1,2,3,4].

This anatomical structure may be the birthplace of glial tumors, which account in this location for 1–5% of all brain tumors [5]. As for glial tumors located in other locations, the ideal treatment is represented by complete surgical resection, followed by adjuvant chemotherapy and/or radiotherapy regimens according to the specific histological grade and international recommendations [6]. Nevertheless, considering the central role that the thalamic area plays in complex functions regulation–most of which remain unknown–and the possible specific motor function activity related to the presence of the corticospinal tract nearby, the possibility of surgical manipulation of this area has remained questionable and highly debated among surgeons for years. Looking at the first available reports, considering the high risk of postoperative morbidity and mortality advocated by many, common thinking witnessed the spreading of conservative treatment strategies based on surgical biopsy followed by adjuvant treatment, with consequent limited impact on postoperative survival [7,8].

As for other pathologies, also the treatment of thalamic tumors benefitted from the progressive development of modern neurosurgery, with the progressive availability of case series reporting better results in terms of postoperative outcomes after resective surgeries in this region [9,10,11,12]. Nevertheless, although promising, the great majority of such works are burdened by small sample sizes and not well-specified methodological workflows. Moreover, such works are often focused on the study of possible surgical approaches or on the natural history of thalamic gliomas rather than specifically analyzing the safety of thalamus violation and the possible survival improvement that a more aggressive surgery may offer to patients. Hence, surgical access to the thalamus remains a case-by-case decision, often based on a multidisciplinary balance regarding patient age, clinical status, and presumed histological diagnosis, among other factors.

The main objective of this work was to analyze a prognosis-based dichotomized (surgery or biopsy) outcome in terms of overall survival (OS), analyzing specific variables that may influence the outcome. A rigorous statistical analysis of different clinical and radiological parameters was performed, also presenting the experience of a third-level neurosurgical Center in Italy, with the aim of better defining surgical indications in this challenging structure.

To the best of our knowledge, this is the first study that evaluated two surgical groups (biopsy or excision surgery), comparing postoperative morbidity and mortality and reporting evidence-based indications for the surgical management of these complex tumors.

## 2. Materials and Methods

The prospectively collected Institutional Pathology Registry, Institutional Complication Registry and Institutional Cancer Registry were queried and retrospectively reviewed to identify all the patients surgically treated at the Fondazione IRCCS “Istituto Neurologico Carlo Besta” between 2006 and 2020 for a thalamic glial lesion. The “Strengthening the Reporting of Observational Studies in Epidemiology (STROBE)” statement and relative guidelines were followed in the exploitation of the study [13]. The possibility of creating and using such databases received the approval of our Institutional Ethical Committee, and informed consent was obtained for all the patients.

### 2.1. Patients Included in the Study and Variables Analyzed

We included in the analysis patients of any age with histologically confirmed diagnoses of primary glial thalamic lesions (according to the 2016 World Health Organization (WHO) classification [14]). The included patients’ data were inserted into an anonymized database, differentiating them based on the surgical procedure performed (biopsy group (BG) or surgery group (SG)).

For all cases, the following variables possibly related to OS were extracted: baseline demographics including age and sex; preoperative, early postoperative and 3-month postoperative Karnofsky Performance Status (KPS) in the adult population and Lansky Performance Status (LPS) in the pediatric one; pre- and post-operative neuro-radiological images along with volumetric segmentation data (preoperative tumor volumes and extent of resection (EOR)); neuropathological data, including tumor grading, immunohistochemistry (IHC) data and genetic results; and OS. Retrospectively until 2014 and prospectively since 2015, the Milan Complexity Scale (MCS) [15], a functional impairment predictive scale in brain tumor surgery, was calculated for every patient: the scale ranges from 0 (less complex cases) to 8 (extremely complex cases) points. Patients for which clinical or follow-up (FU) data were missing were excluded from the analysis.

### 2.2. Pre- and Postoperative Clinical and Radiological Assessments

The clinical and neurological status of patients was evaluated pre- and post-operatively, at the discharge, and with periodical clinical FU every three months, for the first year, in the whole cohort. To note, the first analysis at 3 months was considered necessary to assess the functional recovery and the effectiveness of rehabilitation therapy, overcoming the temporary deficits given by the surgical invasiveness in an extremely complex and eloquent area such as the thalamus. After the first year, FU evaluations were conducted every three months for high-grade gliomas (grades 3 and 4, HGG) and every six months for low-grade gliomas (grades 1 and 2, LGG), as for glial tumors in other areas. Moreover, long-term results in terms of quality of life were assessed by summing information obtained by routine postoperative clinical FU and periodic phone surveys (every 3 months for all the cohort).

From a radiological viewpoint, all patients underwent a complete preoperative magnetic resonance imaging (MRI) study, including T1, T2, FLAIR, contrast-enhanced (CE) sequences, and diffusion tensor imaging (DTI) to reconstruct the corticospinal tract. In selected cases, also computed tomography (CT) and functional-MRI study were performed according to anamnesis and radiological and clinical preoperative data. Postoperatively, a complete neuroradiological work-up was performed for all patients, including early CT (within 24 h after the surgery) and a complete MRI study within 72 h, following Institutional practice and international recommendations [6,16,17]. Long-term radiological FU consisted of MRI imaging every three months for HGG and every 6 months for LGG, as a general rule.

### 2.3. Tumor Segmentation and Volumetric Analyses

Preoperative and postoperative tumor volumes were calculated by a senior neuroradiologist (A.E.) using OsiriX (Pixmeo Sarl, Geneva, Switzerland). The segmentation of the whole contrast enhancement area for HGG or the whole FLAIR hyperintense signal abnormality for LGG on volumetric MRI images was used for tumor volume quantification. The resulting continuous variables were dichotomized into higher or lower preoperative volumes, defined as preoperative tumor volume greater or lesser than the median values originally obtained.

In SG, preoperative and post-operative volumes were compared to calculate the exact EOR according to the formula: [(pre-operative tumor volume–post-operative tumor volume)/preoperative tumor volume] × 100 [18]. Results were categorized into three groups: gross total resection (100% removal-GTR), subtotal resection (90–99% removal-STR) and partial resection (<90% removal-PR).

### 2.4. Surgical Indication and Surgical Procedure

Considering age, preoperative comorbidities, preoperative performance status, the volume of the lesion, and preoperative radiological findings, patients were counseled on the possibility of being submitted to either surgical biopsy or surgical excision of their thalamic lesion. In selected cases, hydrocephalus treatment was proposed as well. The surgical approach was tailored to every patient according to the location and the features of the tumor. For instance, lower age and a lesion strictly confined to the thalamus, without clearly eloquent regions invasion (such as the posterior limb of the internal capsule or the mesencephalic region) were considered surgical candidates, whereas older patients with an already strongly compromised PS were considered more prone for a biopsy procedure, in order to offer the radiotherapy and chemotherapy regimen. It must be added that in our decision-making process, we often consider for maximal safe resection those patients who present with an intact neurologic exam but with a clearly radiological invasion of an eloquent region, as often occurs in younger patients. In fact, our intraoperative neurophysiological data suggest that in such patients, the eloquent function is very often executed by nearby healthy tissue rather than strictly tumoral tissue, offering the possibility of tumor removal from “inside the tumor,” reducing the risk of new-onset postoperative deficits.

Looking at BG, surgical biopsies were performed following common Institutional protocol and international recommendations with a multimodal approach [6], using magnetic neuro-navigation for the selection of the appropriate target (Stealth Station S7, S8–Medtronic—Minneapolis, MN, USA), either with a frameless or stereotactic procedure, according to the surgeon preference. Thanks to the availability of a Pentero900 microscope, all biopsies were performed after administration of sodium fluorescein (SF) at a dosage of 5 mg/Kg, analyzing the degree of SF caption by tissue specimens under YELLOW560 filter (Pentero, Carl Zeiss, Oberkochen, Germany) to further confirm the correctness of the surgical target. In particular, in every case, we observed the biopsy specimens after their collection under YELLOW560 filter activation on a Pentero microscope in order to preliminarily confirm the correctness of the biopsy target. Recent biopsy cases were carried out using confocal laser imaging technology (CONVIVO, Carl Zeiss) with an ex vivo setting to check for tumor cells inside biopsy specimens after their collection. Excisional surgeries were carried out in every case with the use of neuro-navigation (Stealth Station S7, S8) and Intraoperative Neuromonitoring (IONM, Medtronic systems). For every case, somatosensory evoked potentials (SSEP), motor evoked potentials (MEP) and Direct Electrical Cortical Stimulation (DECS) were used. A multidisciplinary conjoined analysis of tumor location, preoperative symptoms, preoperative KPS and preoperative intended objective in terms of the extent of resection led to the choice of the surgical corridor (fronto-orbito-zygomatic craniotomy, anterior interhemispheric transcallosal, anterior contralateral interhemispheric transcallosal, posterior interhemispheric transcallosal, posterior contralateral interhemispheric transcallosal, parieto-occipital, trans-ventricular, and supra-cerebellar infratentorial approaches). We report a sample case of an operated thalamic glioma in Figure 1.

### 2.5. Neuropathological Data and Further Adjuvant Therapies

Neuropathological data obtained from SG and BG were analyzed by an expert neuropathologist (B.P.). Analysis was prospectively made according to the WHO Central Nervous System (CNS) tumor classification existing at the moment of the first histopathological evaluation, and then stored samples were reviewed by a dedicated pathologist (B.P.). For the retrospective part of the study, all the cases were reviewed and reclassified according to the 2016 WHO classification. Biomolecular markers analysis such as IDH-1, p-53, MGMT, PTEN, EGFR, H3 K27M, Trimethyl-H3, ATRX, and PDGFR-α was performed with immunohistochemistry. To note, although the IHC method does not represent the best method to evaluate the methylation status of the promoter of the MGMT gene, the stored samples of GBM were too small to perform genetic analyses in a major part of the patients. In accordance with histological diagnosis, clinical status, and postoperative imaging, the best adjuvant treatment was chosen by a multidisciplinary team consisting of neurosurgeons, neuro-oncologists, and radiotherapists. For LGG, radiological FU without adjuvant treatment was chosen in most cases. For HGG, radiotherapy was used with different protocols of chemotherapy in accordance with the histological findings and most recent literature suggestions [6,19].

### 2.6. Statistical Analysis

Statistical analysis was performed using STATA statistical software, version 15 (StataCorp. 2017. Stata Statistical Software: Release 15. College Station, TX, USA: StataCorp LLC). Comparisons between variables were performed with the use of a t-test or Mann–Whitney test for continuous variables and Chi-square or Fisher exact tests for categorical variables, as appropriate. The Kaplan–Meier method was used to obtain survival curves, survival medians and probabilities at different time points. Log-rank tests were applied to compare survival curves. The Cox proportional hazards models were used to provide hazard ratios (HRs) with the corresponding 95% confidence interval (95% CI) as relative risk estimates of survival. *p* values < 0.05 were considered statistically significant, and all tests were two-sided. To reduce the selection and channeling biases, patients were evaluated by a multidisciplinary team that confirmed the surgical or biopsy indication. All the patients were enrolled after the introduction of the radiotherapy plus concomitant and adjuvant temozolomide protocol. To reduce the heterogeneity of the population cohort, multivariate and subgroup analyses were also performed.

## 3. Results

### 3.1. Patient Demographics, Clinical and Radiological Findings

Overall, 61 patients were identified from January 2006 to December 2020; 56 patients (23 males (41%) and 33 females (59%); F/M ratio = 1.43; mean age 37.7 ± 23.0 years) met the eligibility criteria while five were excluded (three due to extensive involvement of cortico-thalamic regions, one for missing of FU data, and one for a non-surgical related death). Of these, 31 patients (thirteen pediatric subjects (age < 18 years)) were included in SG, whereas 25 patients (three pediatrics) were included in BG. The mean age at diagnosis was 30.0 ± 22.2 years in SG (median 19, range 3–73), while 47.3 ± 20.6 years in BG (median 57, range 3–72, *p* < 0.01).

Overall, the mean tumor volume was 43.5 ± 54.6 mm^3^ with a median volume of 29.2 mm^3^ (range 2.0–338.6 mm^3^). Preoperative volumes in SG and BG were 54.5 ± 69.9 mm^3^ and 30.7 ± 23.6 mm^3^, respectively, *p* = 0.12). In the SG, EOR was reported as GTR in 10 patients (32.3%), STR in 16 patients (51.6%) and PR in five patients (16.1%). Perioperative external ventricular drain (EVD) was needed in six patients (five SG, one BG, *p*= 0.14), ventriculoperitoneal (VPS) shunt in 14 patients (seven in both groups, *p* = 0.64) and third-ventriculostomy (TVS) in seven patients (four in SG, three in BG, *p* = 0.9). MCS was superimposable among the two groups (5.5 ± 1.1; median 5.5, range 4.0–8.0). The median FU period was 104 (interquartile range, IQR, 48–145) months. Perioperative need for hydrocephalus treatment and surgical complications, along with clinical and radiological findings for the entire cohort, are summarized in Table 1.

### 3.2. Neuropathologic Data

Histopathological analysis confirmed the glial nature of all the fifty-six lesions according to the World Health Organization 2016 classification. The population cohort included 11 pilocytic astrocytomas and 20 gliomas (one LGG and 19 HGGs, including eight Diffuse midline gliomas, H3K27M-mutant) in the SG and 25 gliomas (13 LGGs and 12 HGGs, including one Diffuse midline glioma, H3K27M-mutant) in the BG (*p* = 0.01). In five patients (one LGG and four HGG) of the SG, the sample material was limited, and only the distinction between high and low-grade lesions was possible. IDH-1 status was not considered for statistical analysis because it was found negative in all samples. Table 1 summarizes histological data for all patients.

### 3.3. Perioperative and Three-Months Postoperative Performance Status (KPS/LPS Evaluation)

Overall, the mean admission KPS/LPS was 73.2 ± 17.1 (range 40–100); the early postoperative KPS/LPS was 68.5 ± 20.3 (range 20–100), and the 3-month KPS/LPS was 60.7 ± 31.0 (range 0–100).

Considering the two groups separately, the mean admission KPS/LPS score was 75.2 ± 16.5 (range 40–100) in SG and 70.8 ±17.8 (range 40–100) in BG (*p* = 0.35). Early postoperative KPS/LPS was 67.7 ± 20.8 (range 20–100) in SG and 69.6 ± 21.0 (range 30–100) in BG (*p* = 0.74), while 3-month KPS/LPS was 64.5 ± 32.1 (range 0–100) in SG and 56.0 ± 29.7 (range 0–90) in BG (*p* = 0.24; Table 1).

In the SG, the PS decreased in 12 patients (38.7%) and in 11 (35.5%), it was stable. Eight (25.8%) patients experienced an improvement. At 3 months FU, the mean KPS/LPS was slightly reduced if compared to the pre-operative performance status (64.5 ± 32.1; median 70.0, range 0.0–100.0), but only 29.0% of the patients experienced a permanent worsening of the KPS/LPS at three months as compared to preoperative period. Regarding the BG, at discharge, only one patient (4.0%) presented a worsening in the performance status; a slight improvement was observed in one case (4.0%), whereas in the remaining 23 patients (92.0%), the performance status was stable; at 3-months FU a 15 points KPS/LPS mean decrease was noticed (56.0 ± 29.7; median 70.0, range 0.0–90.0) with eight out of 25 patients (32.0%) with a clear worsening of their preoperative KPS/LPS at 3 months. Comparing patients with permanent worsening at 3 months, no statistical difference was noticed between SG and BG (*p* = 0.81).

### 3.4. Outcome Analysis, Overall Survival, and Outcome Predictors

Fourteen patients (12 in SG and 2 in BG) were still alive at the last FU, while six patients (10.7%, three in both the SG and BG) were deceased before the 3-month FU due to tumor progression. Overall, 42 patients died, 19 in the SG and 23 in the BG. Median OS for the entire cohort was 17 months (95% CI, 11–34). A detailed analysis of the variation of OS based on different variables was performed; all the results are reported as Kaplan–Meyer curves in Figure 2.

### 3.5. Overall Survival

Survival at 6, 12, 24 and 60 months were 84%, 74%, 61% and 42% for the SG and 72%, 36%, 23%, and 14% for the BG, respectively (Figure 2A).

### 3.6. Influence of Type of Treatment Received: Surgery vs. Biopsy

The SG showed a better OS compared to the BG, with a median OS of 38 months vs. 10 months, respectively (*p* < 0.01; HR 0.42, 95% CI 0.23–0.78). After covariance adjustment for tumor grade (PA, LGG, and HGG) and patient age (>18, 18–39, 49–59, >60 years), the results showed higher values of OS in the SG compared to BG (*p* = 0.01; HR 0.30, 95% CI 0.12–0.75). The result obtained for the entire group of tumors was maintained considering only HGG and LGG excluding PA (*p* = 0.04; HR 0.35, 95% CI 0.13–0.94). In the HGGs subgroup analysis, a better OS was observed in SG at the univariate analysis (median OS 16 [95% CI, 4–27] months; *p* = 0.01; HR 0.30, 95% CI, 0.12–0.76; Figure 2B) but this association did not remain significant after adjusting for age (*p* = 0.27; HR 0.55, 95% CI, 0.19–1.60). LGG and PA subgroup sample sizes (i.e., one SG vs. 13 BG among LGGs and 11 SG vs. 0 BG among PAs) did not allow any statistical evaluation. At the last FU, the patient submitted to surgery in the LGGs group was alive (32 months), while 12 out of 13 patients submitted to biopsy were dead at the last evaluation (median OS 17, 95% CI 11–35 months).

### 3.7. Influence of Age: Pediatrics vs. Adults

A better prognosis in terms of OS was obtained in the pediatric compared to the adult population (*p* < 0.01; median OS of 65 months [95% CI, 25-NA] vs. 11 months [95% CI, 6–17], respectively). After statistical adjustment considering different tumor grades and different surgical procedures, results were not statistically significant (HR 2.23, 95% CI 0.95–5.23).

### 3.8. Influence of Preoperative Tumor Size

A slight, although not statistically significant, difference was highlighted in the OS of patients (both SG and BG) with higher vs. lower preoperative tumor volumes (relatively to median volume, *p* = 0.36; median OS of 16 months vs. 21 months, respectively; HR 1.34, 95% CI 0.72–2.50).

### 3.9. Influence of EOR

Patients with complete or subtotal resection experienced a median survival time of 38 months (95% CI 13–NA) compared to 10 months (95% CI 6–16) of the BG (*p* < 0.01; HR 0.39, 95% CI 0.20–0.76). This result was not confirmed in multivariate analysis after adjustment for age (*p* = 0.21; HR 0.64, 95% CI 0.31–1.3). Given the expected differences in OS according to histological grade, specific subgroup analyses were conducted. Specifically, in the HGG subgroup, patients who received a GTR/STR of the contrast-enhancing lesion experienced a better OS compared to the BG (*p* = 0.02; median OS of 13 months vs. 5 months, respectively; HR 0.33, 95% CI 0.13–0.85, Figure 2C). This result was not confirmed in multivariate analysis after adjustment for age (*p* = 0.25; HR 0.53, 95% CI 0.17–1.59). Unfortunately, the association of EOR with survival among LGGs and PAs could not be assessed owing to the small number of patients submitted to surgery/biopsy in each group, as mentioned above.

### 3.10. KPS/LPS Analysis

As stated above, at the 3-month evaluation, 29% of SG and 32% of BG patients maintained the PS worsening after surgery without a significant difference between the two groups (*p* = 0.81). We then focused on patients with a stable KPS/LPS at 3 months (39 patients, 22 in SG and 17 in BG); surgical resection (GTR/STR) resulted related to a better OS if compared to biopsy (*p* < 0.01; median OS of 72 months vs. 11 months, respectively; HR 0.31, 95% CI 0.14–0.70, Figure 3A). Notably, OS improvement in the SG remained significant at the multivariate analysis even after age and histological grade adjustment (*p* < 0.01; HR 0.17, 95% CI, 0.05–0.59). A similar association was also found in the HGG population with stable 3-months KPS/LPS (16 patients, 10 SG and six BG): median OS was 21 months in SG and 6 months in BG (*p* < 0.01; HR 0.18, 95% CI 0.04–0.80; Figure 3B). These results were not confirmed in the multivariate analysis (*p* = 0.10; HR 0.25, 95% CI, 0.05–1.29). Survival at 6, 12, and 24 months were 100%, 70%, and 50%, respectively, for the SG, and 83%, 17% and not estimable for the BG. A further analysis, comparing patients with a stable performance status who received a GTR/STR in the SG with patients submitted to biopsy, confirmed the previously observed significant survival advantage in the SG even after age and histological grade adjustment (*p* < 0.01; HR 0.18, 95% CI 0.05–0.63; Figure 3C). Given that patients undergoing surgical resection who maintained a stable 3-month KPS after surgery demonstrated to have remarkable survival advantages over patients undergoing biopsy, we hence analyzed further prognostic factors, finding statistical significance for younger age (*p* = 0.04) and lower histological grade (*p* < 0.01; neither preoperative volume (*p* = 0.32) nor preoperative KPS/LPS (*p* = 0.10) produced significant results). Looking at MCS analysis, in both SG and BG, the mean and median values were 5.5 for both items and both groups.

### 3.11. Influence of Molecular Markers

p53 mutation was statistically associated with the OS: in particular, p53 negativity was associated with a better prognosis (*p* = 0.04; median OS of 8 months vs. 11 months, respectively for p53 positive vs. negative; HR 2.21, 95% CI 1.01–4.82). ATRX gene conservation showed a favorable trend in terms of OS despite a significant result was not reached (*p* = 0.06; median OS of 11 months vs. 29 months, respectively for ATRX positive vs. negative; HR 2.69, 95% CI 0.92–7.83). Looking at HGG only, no statistical significance was reached, although a trend could be found for H3K27M and H3 trimethylation (*p* = 0.08 for both markers). The other molecular markers did not show any other correlation in terms of OS, although those results may be influenced by the limited sample sizes. The IDH status was found as wild type in all the samples. A summary for each molecular marker of the whole group with survival analyses is summarized in Table 2 and Table 3.

## 4. Discussion

Thalamic glial lesions are rare tumors, and their natural history and treatment are only infrequently touched by the neurosurgical literature. Since the thalamus is an important eloquent area with crucial roles in both sensorimotor and superior functions [20], the current neurosurgical attitude is that the treatment must be highly tailored based on the evaluation of preoperative neurological deficits and radiological aspects such as dimension, brainstem involvement and deep veins encasement [12,21].

From a surgical perspective, reaching the thalamus is possible through different corridors and approaches. In 2015 Spetzler and colleagues subdivided the thalamus into six different regions that could be approached through the orbito-zygomatic, ipsilateral trancallosal, contralateral transcallosal, parieto-occipital trans-ventricular, and supracerebellar infrantentorial corridors, that could be addressed via a microsurgical or endoscopic approach [22,23]. Que and colleagues classified unilateral thalamic gliomas into the quadrigeminal cistern and ventricle extension type (Type Q), lateral type (Type L) and anterior type (Type A) according to tumor location, extensive polarity, and location of the ipsilateral posterior limb of the internal capsule, further correlating such clinical-radiological types to different surgical accesses and survivals [24]. More recently, the use of Laser Interstitial Thermal Therapy (LITT) has been advocated as a technical adjunct in the treatment of HGGs, especially for those cases located in difficult and deep areas, such as the thalamus [25,26]. Muray and colleagues, in 2020, presented a case series of 13 consecutive patients treated with LITT for thalamic tumors from 2012 to 2017. Radiographic and clinical characteristics and outcome data were collected, finding this technique as a feasible treatment for patients with such tumors, although more studies comparing treatment modalities of thalamic tumors were advocated [25].

Recent literature has taken into account the existence of different prognostic factors (i.e., age, MGMT methylation status, EOR, preoperative PS) that may variably influence OS in gliomas, regardless of their location [6]. Looking at this specific aspect, Palmisciano and colleagues recently reviewed 25 studies comprehending 617 patients affected by thalamic gliomas, finding that adult thalamic gliomas, especially the ones with the H3 K27 mutation, are associated with poor survival and that complete surgical resection is associated with improved survival rates but is not always feasible [27]. Looking at the existing literature, the relative importance of such prognostic factors remains to be determined. Moreover, the feasibility of obtaining a GTR in the thalamic area has not been clarified in detail, and only a few studies investigated the role of surgical resection compared to biopsy in patients with a new presumptive diagnosis of thalamic glial lesions, reaching conflicting results [28,29].

In the present study, we directly compared two cohorts of patients, either submitted to surgical removal or biopsy procedure for thalamic glial lesions. The two cohorts were compared in terms of OS and PS. Our findings suggested acceptable postoperative outcomes that appear comparable to those seen in historic controls of operated supra-tentorial lobar gliomas [7]. Focusing on thalamic glioma surgery, our findings were comparable to the literature [9,10,30].

In the present series, a better OS was observed in the SG compared to the BG considering the entire population, also after adjustment per age and histological grade. Similar results were found when considering the diffuse glioma subgroup alone, in accordance with current literature [31,32].

Looking at EOR, its significance as a positive prognostic factor in terms of OS in gliomas is well known [18,33]. Accordingly, we observed a significant OS advantage in patients submitted to GTR/STR surgery rather than biopsy. The specific sub-analysis in the HGG group confirmed a significant advantage in OS following surgical resection. Although not confirmed by multivariate analysis, this favorable trend showed a favorable hazard ratio, which would have needed a larger sample size to achieve statistical significance. Hence, we demonstrated the significance of obtaining a GTR/STR even in an extremely challenging and eloquent area such as the thalamus. Unfortunately, the small sample of thalamic LGG analyzed did not permit us to draw any significant conclusion regarding the role of EOR [34,35].

Concerning the negative correlation between preoperative tumor volume and OS, we noticed a worse prognosis in parallel to the increase of preoperative tumor volume, i, without also reaching statistical significance in this subgroup. Moreover, the specific anatomical location and spreading of the tumor may correlate with the prognosis apart from a mere volumetric analysis. In fact, OS in thalamic gliomas with different anatomical extensions, for instance, in tumors causing hydrocephalus or with hypothalamic/brainstem involvement, may be very different. This aspect would deserve further study with a larger sample size that would permit specific subgroup analyses.

It is now clear that molecular markers owe prognostic significance in glioma patients. Many reports are available about HGG molecular profiling and its implication on OS [36], although very few are focused on thalamic HGG, and very often, statistical analysis is not reported [9]. Considering the entire cohort of this study, our data highlighted the negative prognostic value in terms of OS in the case of p53 gene mutations. A similar trend was found, considering the absence of ATRX gene mutation, without reaching statistical significance. According to previous papers, and also in our cohort, the EGFR gene amplification was linked to a worse prognosis [37], with a trend that was not significant. As for EGFR, a significant survival advantage between PTEN, PDGFR-α status and variations in the expression of the MGMT protein and OS was not detected. Regarding the MGMT analysis, theoretically, the low expression of the MGMT protein, as detachable from IHC, should be related to the methylation of the promoter of the MGMT gene, which is a positive prognostic factor [36]. In recent years, this topic has been highly debated, and modern studies showed a frequent discordance between MGMT expression as detected by IHC and by MGMT DNA methylation [38]; therefore, the authors considered that IHC regarding the MGMT protein should not be considered as a survival marker. Finally, the histone H3 gene was analyzed. H3 K27M mutation is related to a specific subgroup of diffuse midline gliomas, which are typically age- and site-dependent, localizing in the midline structures of children and young adults and associating with the worst prognosis [39,40]. Looking at H3 gene results, although following a trend that could be interpreted as inverted with respect to current literature (*p* = 0.08), they should be considered as a simple survival analysis that may be influenced by all the other factors, such as surgical group, age, EOR etc. Moreover, the low number of patients that were found to have such mutations does not permit us to make any definitive conclusion on the topic.

KPS/LPS was used for analyzing the PS of the patients in this study. The correlation between patients’ PS and the impact of surgery on it recently became a trending topic in the neurosurgical literature. In fact, recent studies showed the preoperative KPS/LPS as an independent factor for a better OS in glioma patients [9,10,30]. In this regard, to the best of our knowledge, this is the first study that analyzed in detail the impact of the operative treatment on the postoperative KPS/LPS in both the short and long term in thalamic glioma patients.

Concerning this critical point, at first, we noticed a decrease in the KPS/LPS indexes in the early postoperative period of patients in SG (38.7%) rather than in the BG (4%), as expected, considering the major invasiveness of surgical removal. Surprisingly, a reversal of the trend occurred in the following period: from 4% to 32% in BG and from 38.7% to 29.0% in SG at 3 months. We consider this trend inversion in the case of surgical patients a consequence of the reduction of the mass effect and perilesional edema due to tumor removal and a worsening of the same variables in patients undergoing biopsies, resulting in a slow but inexorable clinical deterioration. Furthermore, analyzing patients with unchanged KPS/LPS at the 3-month FU in the entire population, a better OS was found in the SG at the univariate and multivariate analysis, also after correction for age and histological grade. The results appeared as an interesting topic, especially considering the HGG subgroup. In fact, in HGG tumors that maintained stable KPS/LPS at three months postoperatively, we found a better prognosis (median OS of 21 months in SG vs. 6 months in BG). This result was not confirmed in the multivariate analysis. Nevertheless, the analysis of the subgroup of patients with a stable KPS/LPS at 3 months that underwent GTR/STR confirmed longer survival for SG, also after adjustment for age and histological grade. In other words, the common attitude of considering surgery for eloquent and deep-seated lesions as often burdened by a high risk of morbidity may be re-discussed [41,42,43].

Considering the strong survival advantage found in those patients undergoing surgical resection who maintained a stable 3-month KPS after surgery, we finally analyzed further prognostic factors for having a stable 3-month postoperative KPS/LPS, as these patients were shown to be a unique cohort of glioma patients that may benefit most from surgical resection. In our series, statistical significance was found for younger age and lower histological grade, with a strong positive trend for lower preoperative tumor volumes and higher preoperative KPS/LPS.

Concerning this aspect, a few years ago, our group proposed the MCS as a useful and simple tool that may provide prognostic information starting from the analysis of perioperative clinical and radiological data [15]. In this specific case, MCS appeared inadequate in classifying thalamic gliomas, due to the substantial complete eloquence of the region, with common high tumor volumes or venous encasement. As a matter of fact, regarding the SG, the MCS range was 4–8 points in all cases, which made it little useful to predict functional impairment. Therefore, new parameters are needed to elaborate a preoperative prognostic scale for thalamic HGG.

In conclusion, taken together, all these data seem to highlight the role of surgery also in the treatment of thalamic HGG. In particular, surgery has been demonstrated to offer a strong survival advantage when tumor removal is attempted and STR/GTR is obtained. Nevertheless, as outlined by our data, it seems to be essential to try to identify preoperative favorable prognostic factors for a good postoperative recovery.

This study is limited by its retrospective design and the intrinsic selection bias. In fact, due to the extreme case-by-case surgical indication that is given for pathology in this area, patients were obviously not randomly assigned to biopsy or surgical procedure, which meant that the treatment might have been biased by the multidisciplinary team preference based on the preoperative condition of the patient and features of the tumor. Additionally, the heterogeneity of the patients analyzed in the present study resulted in subgroup cohorts with limited numbers, a condition that contrasts with the rarity of the pathology. Moreover, the molecular profiling availability only for patients affected by GBM limits our molecular considerations.

Another aspect that should be outlined is that this work analyzed patients collected over a very long period (2006–2020). In such a period, almost three classifications of CNS tumors by the WHO were edited. This aspect could affect the generalizability of our findings, given that tumors that in one edition were considered as a subclassification could be found in another subclassification in a different edition. Moreover, only in the last 10 years has the WHO outlined the importance of molecular classification in relation to survival. Hence, although we re-read all our histological findings from the older cases, this intrinsic limitation could not be overcome. Further studies, possibly associating more centers with more recent cases, considering the rarity and the still debated management of this region, are necessary to define the best management approach for thalamic gliomas.

## 5. Conclusions

This study evaluated a large mono-institutional cohort of patients with thalamic gliomas where surgery was demonstrated to offer a stronger survival advantage when tumor removal is attempted and STR/GTR is obtained. Aiming to improve quality of life and OS, a precise evaluation of predictors of the 3-month postoperative PS was found to be crucial. Considering our data, statistical significance was found for lower age and histological grade, with a strong positive trend for lower preoperative tumor volumes and higher preoperative KPS/LPS. Further prospective, multi-center studies are needed to better elucidate prognostic factors for thalamic gliomas, especially considering those aspects that may correlate with a good postoperative recovery that was shown to be strictly linked to an improvement in OS and quality of life.

## Figures and Tables

**Figure 1 cancers-15-00361-f001:**
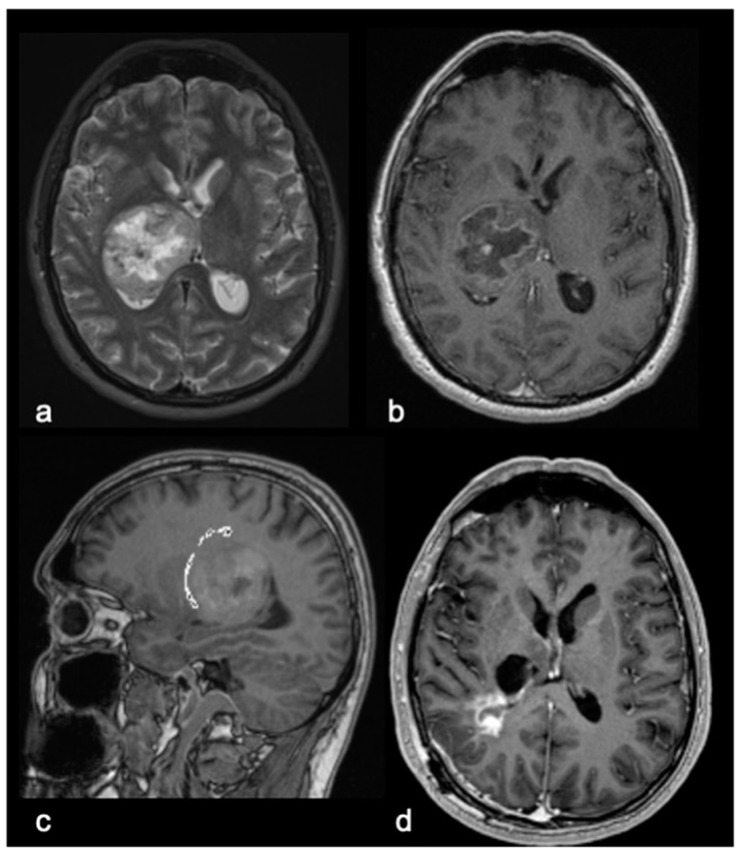
An illustrative case of a 51-year-old right-handed male with a 3-month history of rapidly progressive right sensorimotor syndrome. (**a**). Axial T2-weighted Brain Magnetic Resonance imaging (MRI). (**b**). Axial postcontrast T1-weighted MRI and Sagittal T1-weighted MRI with tractographic reconstruction (**c**) of the right corticospinal tract, demonstrating a voluminous ring-enhancing right thalamic lesion. The patient underwent surgical removal of the lesion through a right trans-parietal approach. (**d**). Axial postcontrast T1-weighted MRI showing subtotal removal of the lesion. Postoperatively, the patient presented a transient worsening of right hemiparesis with recovery through rehabilitation after about 1 month. Histological examination confirmed an IDH wild-type glioblastoma. The patient underwent adjuvant therapy with Temozolomide and concomitant Radiotherapy, Regorafenib and Lomustine, with a survival of 18 months.

**Figure 2 cancers-15-00361-f002:**
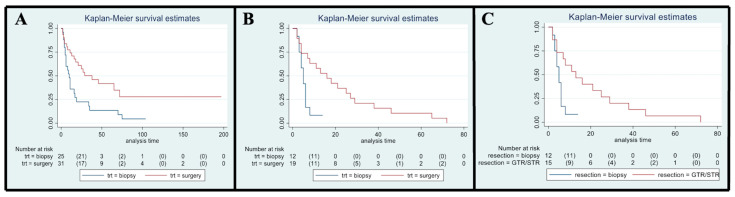
Kaplan–Meyer curves. (**A**). OS in all patients comparing surgery and biopsy. (**B**). Univariate analysis in the HGG subgroup. (**C**). OS considering the extent of resection in the HGG sub-group.

**Figure 3 cancers-15-00361-f003:**
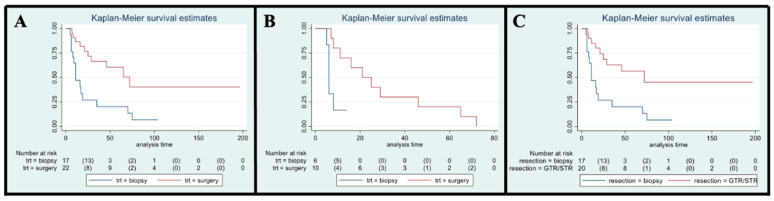
(**A**). OS in stable KPS/LPS at 3 months. (**B**). OS in stable KPS/LPS at 3 months in the HGG subgroup. (**C**). OS in stable KPS/LPS at 3 months considering GTR/STR vs. biopsy.

**Table 1 cancers-15-00361-t001:** Demographics, histological diagnosis and tumor volumes.

Variables	Surgery Group (SG, n 31)	Biopsy Group (BG, n 25)	*p*-Value *
Sex			
Male	13 (41.9%)	10 (40.0%)	*p* = 0.88
Female	18 (58.1%)	15 (60.0%)	
Age at diagnosis			
Mean (SD), y	30.0 (22.2)	47.3 (20.6)	
Median (range), y	19 (3–73)	57 (3–72)	*p* < 0.01
Adults	18 (58.1%)	22 (88.0%)	*p* = 0.01
Pediatrics	13 (41.9%)	3 (12.0%)	
Histological diagnosis			
Other astrocytic tumors	11 (35.5%)	0 (0.0%)	*p* < 0.01
Pilocytic astrocytoma	11	0	
WHO grade II	1 (3.2%)	3 (12.0%)	
astrocytoma, IDH WT	1	2	
LGG	0	1	
WHO grade III	0 (0.0%)	10 (40.0%)	
Anaplastic astrocytoma, NOS	0	7	
Anaplastic oligodendroglioma, NOS	0	1	
WHO grade IV	19 (61.3%)	12 (48.0%)	
Glioblastoma, IDH WT	11	3	
Glioblatoma, NOS	0	4	
Diffuse midline glioma, H3K27M-mutant	9	1	
HGG	0	4	
Tumor Volume at diagnosis (mm^3^)			
Mean (SD)	54.5 (69.9)	30.7 (23.6)	
Median (range)	32.2 (2.0–338.6)	23.6 (5.1–107.2)	*p* = 0.12
Milan Complexity Scale (MCS) score at diagnosis			
Mean (SD)	5.5 (1.1)	5.5 (1.1)	
Median (range)	5.5 (4–8)	5.5 (4–8)	*p* = 1.00
Perioperative EVD, VPS, and TVS			
EVD	5 (16.1%)	1 (4.0%)	*p* = 0.14
VPS	7 (22.6%)	7 (28.0%)	*p* = 0.64
TVS	4 (12.9%)	3 (12.0%)	*p* = 0.92
*Perioperative complications* (overall) CSL leak Infections	5 (16.1%) 3 (9.6%) 2 (6.5%)	2 (8.0%) 1 (4.0%) 1 (4.0%)	*p* = 0.45 *p* = 0.75 *p* = 0.85
KPS at admission			
Mean (SD)	75.2 (16.5)	70.8 (17.8)	*p* = 0.35
Median (range)	80 (40–100)	70 (40–100)	
KPS at discharge			
Mean (SD)	67.7 (20.8)	69.6 (21.0)	*p* = 0.74
Median (range)	70 (20–100)	70 (30–100)	
KPS at 3 months			
Mean (SD)	64.5 (32.1)	56.0 (29.7)	*p* = 0.24
Median (range)	70 (0–100)	70 (0–90)	

* *p*-values from *t*-test or Mann-Whitney test for continuous variables, and Chi-square or Fisher exact test for categorical variables, as appropriate.

**Table 2 cancers-15-00361-t002:** Molecular characterization of LGG and HGG with survival estimates. SG and BG samples appear as a unique cohort to reduce the impact that different treatments would have manifested on the results produced.

GLIOMAS					
Biomarker	Number of Patients	Median OS (Months)	*p*-Value (Log-Rank Test)	HR (95% CI)	Kaplan-Meyer Curves
p-53	42				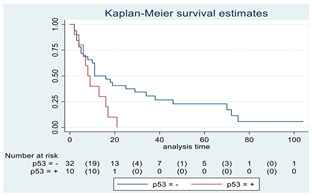
Positive	10	8	0.04	2.21 (1.01–4.82)	
Negative	32	11			
MGMT	15				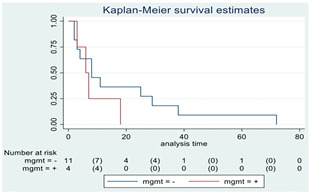
Positive	4	6	0.29	1.92 (0.55–6.70)	
Negative	11	8			
PTEN	16				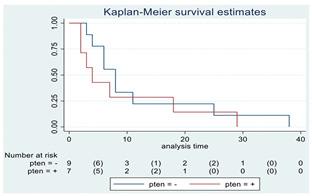
Positive	7	4	0.41	1.51 (0.54–4.21)	
Negative	9	8			
EGFR	19				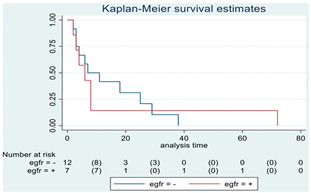
Positive	7	6	0.82	1.12 (0.41–3.08)	
Negative	12	7			
H3-K27M	21				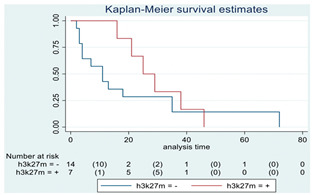
Positive	7	25	0.22	0.54 (0.19–1.50)	
Negative	14	11			
H3 trimethylation	21				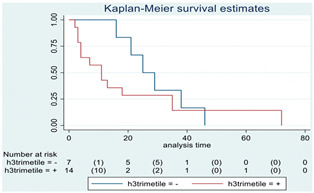
Positive	14	11	0.22	1.85 (0.67–5.14)	
Negative	7	25			
ATRX	16				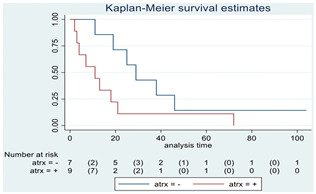
Positive	9	11	0.06	2.69 (0.92–7.83)	
Negative	7	29			
PDGFR-α	12				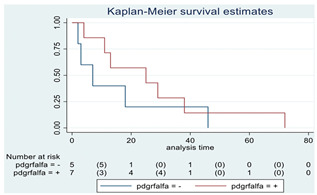
Positive	7	25	0.35	0.57 (0.17–1.89)	
Negative	5	7			
IDH	56				
Positive	0	N/A	N/A	N/A	N/A
Negative	56	17			

Abbreviations: CI, confidence interval; HR, hazard ratio; OS, overall survival.

**Table 3 cancers-15-00361-t003:** Molecular characterization of HGG with survival estimates. SG and BG samples appear as a unique cohort to reduce the impact that different treatments would have manifested on the results produced.

HGG					
Biomarker	Number of Patients	Median OS (Months)	*p*-Value (Log-Rank Test)	HR (95% CI)	Kaplan-Meyer Curves
p-53	26				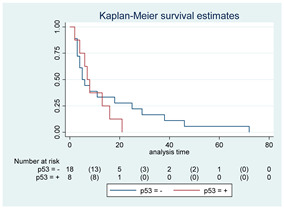
Positive	8	7	0.52	1.33 (0.55–3.26)	
Negative	18	5			
MGMT	15				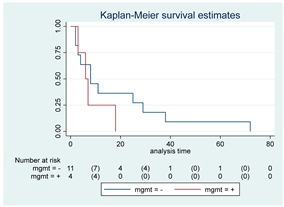
Positive	4	6	0.30	1.92 (0.55–6.70)	
Negative	11	8			
PTEN	16				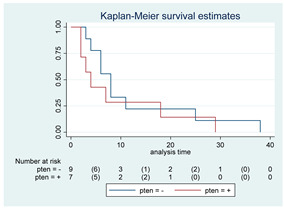
Positive	7	4	0.43	1.50 (0.54–4.20)	
Negative	9	8			
EGFR	19				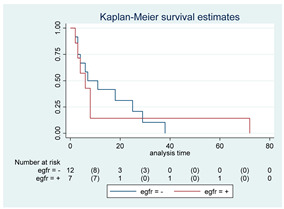
Positive	7	6	0.82	1.12 (0.40–3.07)	
Negative	12	7			
H3-K27M	17				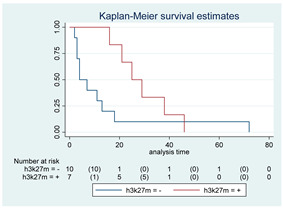
Positive	7	25	0.08	0.38 (0.13–1.13)	
Negative	10	4			
H3 trimethylation	17				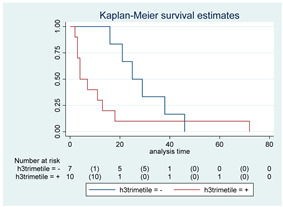
Positive	10	4	0.08	2.59 (0.88–7.62)	
Negative	7	25			
ATRX	13				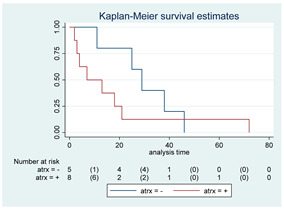
Positive	8	7	0.29	1.88 (0.57–6.20)	
Negative	5	29			
PDGFR-A	12				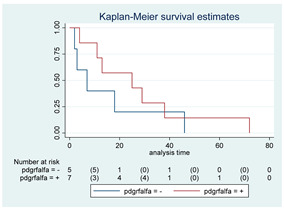
Positive	7	25	0.35	0.56 (0.16–1.88)	
Negative	5	7			

Abbreviations: CI, confidence interval; HR, hazard ratio; OS, overall survival.

## Data Availability

Data are available upon request to the corresponding author.

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
