# Peer review of "Are Thalamic Intrinsic Lesions Operable? No-Man’s Land Revisited by the Analysis of a Large Retrospective, Mono-Institutional, Cohort"

_cancers, 2023, doi:10.3390/cancers15020361_

Round 1

Reviewer 1 Report

In the manuscript "Are thalamic intrinsic lesions operable? No-man's land revisited by the analysis of a large retrospective, mono-institutional, cohort " Paolo Ferroli and colleagues report the result of a monocentric, retrospective study on 61 patients affected by newly thalamic glioma, submitted to either biopsy or resection of the tumor. Both pediatric and adult subjects were included in the study. Patients were divided in two groups one submitted to Surgery Group (SG, median age 19) and one Biopsy (BG, median age 57). Retrospective statistical analysis of their data indicated that patients with a stable 3-months KPS after surgery have a better prognosis in terms of OS compared to biopsy. Moreover, as it is often the case with gliomas, the authors confirmed that age at diagnosis was inversely related to survival. 

The idea that thalamic gliomas, that are surgically more demanding, may benefit from extensive resection is, as indicated also by the authors, not new. However, their surgical case series may stimulate a larger interest for thalamic glioma resection in the neurosurgical community.

However, I see many problems in supporting their conclusions that " surgery demonstrated to be more than a safe alternative to biopsy, offering strong survival advantage when tumor removal is attempted".

Thalamic gliomas are usually lumped together with other midline gliomas that are often associated with histone genes mutations and have a 5-year survival of approximately 48% or even better if only pediatric patients are considered. The tumor histotypes listed in the present study are very heterogeneous and their distribution in the two groups of patients not balanced. Moreover, the median age of the patients submitted to biopsy is significantly higher compared to that of the patients submitted to surgical resection. These obvious biases, that presumably reflect a careful selection by the surgeons before proceeding with surgery or biopsy on a thalamic tumor makes less convincing the authors conclusions. The addition of a more careful discussion of the limitations of the study will certainly improve the manuscript.

Minor points:

1) The true accrual period is unclear: in the abstract the authors indicate 2006-2020 while in the Materials and Methods and Results section they indicate 2010-2020. Please clarify.

2) The number of citations related to thalamic surgery is rather limited. Two recent additions that should be considered and discussed are the following:

Que T, Li Z, Zheng H, Tan JE, Yuan X, Yi G, Fang L, Nie J, Yin Y, Xu H, Zheng X, Liu J, Zhang XA, Qi S, Huang G. Classification of unilateral thalamic gliomas predicts tumor resection and patient's survival: a single center retrospective study. J Neurosurg Sci. 2022 Apr 13. doi: 10.23736/S0390-5616.22.05660-0. 

Liu S, Xie T, Wu S, Li C, Liu T, Zhao P, Chen P, Zhang X. Endoscopic resection of thalamic lesions via supracerebellar infratentorial approach: a case series and technical note. Neurosurg Rev. 2022 Dec;45(6):3817-3827. doi: 10.1007/s10143-022-01891-4. 

Author Response

Reviewer 1

In the manuscript "Are thalamic intrinsic lesions operable? No-man's land revisited by the analysis of a large retrospective, mono-institutional, cohort " Paolo Ferroli and colleagues report the result of a monocentric, retrospective study on 61 patients affected by newly thalamic glioma, submitted to either biopsy or resection of the tumor. Both pediatric and adult subjects were included in the study. Patients were divided in two groups one submitted to Surgery Group (SG, median age 19) and one Biopsy (BG, median age 57). Retrospective statistical analysis of their data indicated that patients with a stable 3-months KPS after surgery have a better prognosis in terms of OS compared to biopsy. Moreover, as it is often the case with gliomas, the authors confirmed that age at diagnosis was inversely related to survival.

The idea that thalamic gliomas, that are surgically more demanding, may benefit from extensive resection is, as indicated also by the authors, not new. However, their surgical case series may stimulate a larger interest for thalamic glioma resection in the neurosurgical community.

However, I see many problems in supporting their conclusions that " surgery demonstrated to be more than a safe alternative to biopsy, offering strong survival advantage when tumor removal is attempted".

Thalamic gliomas are usually lumped together with other midline gliomas that are often associated with histone genes mutations and have a 5-year survival of approximately 48% or even better if only pediatric patients are considered. The tumor histotypes listed in the present study are very heterogeneous and their distribution in the two groups of patients not balanced.

Answer: We thank the reviewer for this valuable comment. The answer to this smart observation finds its rationale in the very long period in which we collected all these patients (2006-2020). In such a period, almost three classification of CNS tumors by WHO were created. The importance of midline gliomas with H34 histone modifications, especially in pediatric patients, has been added just in the previous version of the classification, but it was not present in the 2016 Edition. Considering this “intrinsic” limitation of our study (that was added to the limitations section), we just tried to analyze as a whole the entire cohort, knowing that every classification we could use it would have been not completely appropriate from a “classification” point of view, and of course also from a “survival” point of view, due to the different survival advantages that the reviewer correctly outlined for some specific types of gliomas, molecularly defined. Nevertheless, we considered acceptable this limitation when creating this work because, as a matter of fact, still few series with so many patients are present in literature, and very often in such works a precise analysis of prognosticators is not performed. We feel that, keeping in mind the limitation raised by the reviewer, the readers may anyway obtain interesting data from this work. Anyway, we modified our “conclusion” sentence, as suggested by the reviewer.

Moreover, the median age of the patients submitted to biopsy is significantly higher compared to that of the patients submitted to surgical resection. These obvious biases, that presumably reflect a careful selection by the surgeons before proceeding with surgery or biopsy on a thalamic tumor makes less convincing the authors conclusions.

Answer: We thank the reviewer for this comment. As outlined in the limitations section, one of the most important limits of our work is exactly the one raised by the reviewer. In lines 497-501 we wrote: “In fact, due to the extreme case-by-case surgical indication that is given for a pathology in this area, patients were obviously not randomly assigned to biopsy or surgical procedure, which meant that the treatment might have been biased by the multidisciplinary team preference based on the preoperative condition of the patient and features of the tumor.”. Nevertheless, we can say that every effort was performed to lower this negative aspect, as we performed various analysis also adjusting them per age, among other factors. For sure this aspect could not completely overcome the issue, that should be interpreted as an “intrinsic selection bias” linked to the nature of the study.

The addition of a more careful discussion of the limitations of the study will certainly improve the manuscript.

Answer: We thank the reviewer for the question. As requested, we proceeded with an enrichment of the limitations section at the end of discussion section, following her/his and other reviewers’ suggestions.

Minor points:

  • The true accrual period is unclear: in the abstract the authors indicate 2006-2020 while in the Materials and Methods and Results section they indicate 2010-2020. Please clarify.

Answer: Sorry for this typing mistake. The period of study is, actually, 2006-2020. We modified this aspect through the entire manuscript.

  • The number of citations related to thalamic surgery is rather limited. Two recent additions that should be considered and discussed are the following:

Que T, Li Z, Zheng H, Tan JE, Yuan X, Yi G, Fang L, Nie J, Yin Y, Xu H, Zheng X, Liu J, Zhang XA, Qi S, Huang G. Classification of unilateral thalamic gliomas predicts tumor resection and patient's survival: a single center retrospective study. J Neurosurg Sci. 2022 Apr 13. doi: 10.23736/S0390-5616.22.05660-0.

Liu S, Xie T, Wu S, Li C, Liu T, Zhao P, Chen P, Zhang X. Endoscopic resection of thalamic lesions via supracerebellar infratentorial approach: a case series and technical note. Neurosurg Rev. 2022 Dec;45(6):3817-3827. doi: 10.1007/s10143-022-01891-4.

Answer: We thank the reviewer for this comment. As requested, we further expanded the parts of the text and citations related to thalamic surgery, as requested.

Reviewer 2 Report

These authors review their experience with surgical management of thalamic tumors.

1)    This manuscript would benefit from review for English language.

2)    The authors acknowledge that surgery for tumors of the thalamus can be treacherous due to possible impacts on thalamic function, some of which remain unknown.  However, surgery for thalamic tumors can also lead to damage to surrounding structures such as the internal capsule, leading to predictable postoperative deficits.

3)    In the abstract, the authors indicate that the database was queried for patients undergoing surgery from 2006-2020.  In the methods section, they report the inclusion years as 2010-2020.  Please correct this discrepancy.

4)    Please clarify the biopsy procedure.  The authors indicate that fluorescein and the operating microscope were used for biopsies, suggesting that these procedures were performed open.  However, they also refer to the use of frameless or frame-based stereotactic systems which, while they are often used for open craniotomies, are also commonly used for stereotactic needle biopsies, especially in deep locations such as the thalamus.

5)    Please define LGG vs HGG.  Presumably LGG was WHO grade 1-2 and HGG was grade 3-4?

6)    The authors report “11 pilocytic astrocytoma and 20 gliomas.”  As the authros know, pilocytic astrocytomas are glial tumors.  However, the phrasing should be clarified in the manuscript.

7)    The Kaplan-Meyer curves in Figure 2 for mixed histologies (A, C, D, E) are less useful than they curves demonstrating only HGG (B, F).  The mixed histology curves are significantly confounded since the two groups are not otherwise matched.

8)    The H3K27M mutation analysis as presented in Table 3 seems to indicate that patients harboring the mutation had a survival advantage.  This is at odds with the prevailing literature.  Are these patients who meet all diagnostic criteria for H3K27M mutated diffuse midline gliomas, or does it include all patients with the mutation regardless of whether they meet the diagnostic criteria for this particularly devastating tumor.

9)    As this is a surgical series, the authors should at least briefly report on surgical complications such as infection, CSF leak, neurological deficit, etc.

Author Response

Reviewer 2

These authors review their experience with surgical management of thalamic tumors.

  • This manuscript would benefit from review for English language.

Answer: Thanks for the suggestion, we proceeded with a complete English review by a mother-tongue speaker.

  • The authors acknowledge that surgery for tumors of the thalamus can be treacherous due to possible impacts on thalamic function, some of which remain unknown. However, surgery for thalamic tumors can also lead to damage to surrounding structures such as the internal capsule, leading to predictable postoperative deficits.

Answer: This is completely true, and we thank the reviewer for having pointed out such aspect. We underlined it in the Introduction section.

  • In the abstract, the authors indicate that the database was queried for patients undergoing surgery from 2006-2020. In the methods section, they report the inclusion years as 2010-2020.  Please correct this discrepancy.

Answer: Sorry for this typing mistake. As answered also to the first reviewer, the period of study is, actually, 2006-2020. We modified this aspect through the entire manuscript. Sorry again.

  • Please clarify the biopsy procedure. The authors indicate that fluorescein and the operating microscope were used for biopsies, suggesting that these procedures were performed open.  However, they also refer to the use of frameless or frame-based stereotactic systems which, while they are often used for open craniotomies, are also commonly used for stereotactic needle biopsies, especially in deep locations such as the thalamus.

Answer: Sorry for the misunderstanding, we did not clarify enough the procedure in the manuscript. We usually perform biopsies procedures either open or frame-less with the aid of navigation systems. We usually perform both after the administration of 5 mg/Kg of sodium fluorescein at patient intubation. This aspect obviously helps a lot during open procedures in recognizing the tumor, but very often helps during frameless procedures as well, as the collected tumor specimens may be seen under YELLOW 560 visualization on Pentero, preliminary confirming or not the presence of tumor tissue in the specimens obtained. We clarified such aspect in the manuscript.

  • Please define LGG vs HGG. Presumably LGG was WHO grade 1-2 and HGG was grade 3-4?

Answer: Thanks for the question. It is correct, we identified such macro-categories to simplify the statistical analysis and the generalizability of our results, considering that during the very long period analyzed (patients operated on between 2006 and 2020) almost three classification of CNS tumors were created.

  • The authors report “11 pilocytic astrocytoma and 20 gliomas.” As the authros know, pilocytic astrocytomas are glial tumors.  However, the phrasing should be clarified in the manuscript.

Answer: Thanks for the observation. We decided to keep separated such entities due to the major prevalence of such tumor in the pediatric population, and due to the fact that the survival estimates are usually very different between gliomas (generally speaking) and pilocytic astrocytomas.

  • The Kaplan-Meyer curves in Figure 2 for mixed histologies (A, C, D, E) are less useful than they curves demonstrating only HGG (B, F). The mixed histology curves are significantly confounded since the two groups are not otherwise matched.

Answer: Thanks for the suggestion, we modified the Figure as requested, leaving just OS for the entire cohort (A) and the KM curves for HGG (B, F). We modified the text accordingly.

  • The H3K27M mutation analysis as presented in Table 3 seems to indicate that patients harboring the mutation had a survival advantage. This is at odds with the prevailing literature.  Are these patients who meet all diagnostic criteria for H3K27M mutated diffuse midline gliomas, or does it include all patients with the mutation regardless of whether they meet the diagnostic criteria for this particularly devastating tumor.

Answer: Thanks for the question. As outlined, the results, although following a trend that could be interpreted as inverted in respect to current literature, should be considered as a mere survival analysis. Firstly, the result is not statistically significant. Secondly, this was just a simple genetic analysis and, given that during the very long period analyzed (patients operated on between 2006 and 2020) almost three classification of CNS tumors were created, we decided to include in this subgroup all patients with the mutation regardless of whether they meet the diagnostic criteria. Thirdly and lastly, such data may be influenced by all the other factors, for instance a major part of the H3K27M positive patients may be surgical patients, in which just the surgical procedure itself may be the main reason for the survival benefit. Moreover, the low number of patients that were found to have such mutation does not permit to make any definitive conclusion on the topic. Hence, such data may be interpreted cautiously and should be interpreted as not statistically significant data, reason for which they were poorly discussed in the appropriate section. Nevertheless, we added a brief comment on such aspect to increase clarity of data analysis.

  • As this is a surgical series, the authors should at least briefly report on surgical complications such as infection, CSF leak, neurological deficit, etc.

Answer: Thanks for the point. It is correct and we are sorry for not having included them it in the previous version. We added CSF leaks and infections as major categories, calculating also an overall rate of infections for the two groups. Hydrocephalus rate was already reported. To simplify, we did not insert “new neurological deficits” as this aspect may be indirectly evaluated through the combination of pre and postoperative Performance Status, which are the main indicators in our work. We added in Table 1 the complication report for our cohort, as requested.

Reviewer 3 Report

The authors present a single institution retrospective analysis focused on the surgical management and outcomes of patients with thalamic gliomas.
Although this is a well written, recent reviews on this have recently been done in a recent study published last year by Palmisciano et al. (1), thus this study does not add much to the literature.
Additionally, thalamic lesions can radiographically be in different areas of the thalamus and have varying involvement of surrounding structures, pushing the thalamic nuclei medial, lateral, post/anterior. Therefore approach largely depends on where bulk of lesion may be. Authors should further
The discussion on the use of LITT for these tumors is interesting and should be discussed.
Authors should also review paper for grammar, spelling, and formatting. 

(1) Palmisciano P, El Ahmadieh TY, Haider AS, Bin Alamer O, Robertson FC, Plitt AR, Aoun SG, Yu K, Cohen-Gadol A, Moss NS, Patel TR, Sawaya R. Thalamic gliomas in adults: a systematic review of clinical characteristics, treatment strategies, and survival outcomes. J Neurooncol. 2021 Dec;155(3):215-224.

Author Response

Reviewer 3

The authors present a single institution retrospective analysis focused on the surgical management and outcomes of patients with thalamic gliomas.

Although this is a well written, recent reviews on this have recently been done in a recent study published last year by Palmisciano et al. (1), thus this study does not add much to the literature. Additionally, thalamic lesions can radiographically be in different areas of the thalamus and have varying involvement of surrounding structures, pushing the thalamic nuclei medial, lateral, post/anterior. Therefore approach largely depends on where bulk of lesion may be.

(1) Palmisciano P, El Ahmadieh TY, Haider AS, Bin Alamer O, Robertson FC, Plitt AR, Aoun SG, Yu K, Cohen-Gadol A, Moss NS, Patel TR, Sawaya R. Thalamic gliomas in adults: a systematic review of clinical characteristics, treatment strategies, and survival outcomes. J Neurooncol. 2021 Dec;155(3):215-224.

Answer: We thank the reviewer for this valuable comment. We added this work to the reference list and we also discussed it in the appropriate section. Nevertheless, in our opinion, what we can say as a conclusion in our work is slightly different from what can be learnt in this review. In their work, Palmisciano and colleagues conclude that “Adult thalamic gliomas are associated with poor survival. Complete surgical resection is associated with improved survival rates but is not always feasible. H3 K27M mutation is associated with worse survival and a more aggressive approach should be considered for mutant neoplasms.” None of such data is un-waited considering thalamic glioma surgery. We, instead, demonstrated that a strong survival advantage was found in those patients undergoing surgical resection who maintained a stable 3-months KPS after surgery, further analyzing eventual prognosticators for this specific condition, meaning that it seems to be essential trying to identify preoperative favorable prognostic factors for a good postoperative recovery, as outlined in the conclusions. Nevertheless, as the work of Palmisciano and colleagues is a very well-done work, could adding scientific value to our manuscript, we added it to our references.

Authors should further The discussion on the use of LITT for these tumors is interesting and should be discussed.

Answer: Correct point, we added a brief discussion on such aspect, adding relative references as well.

Authors should also review paper for grammar, spelling, and formatting.

Answer: Thanks for the observation, we went through a complete revision by a mother-tongue speaker.

Round 2

Reviewer 3 Report

Revisions have been acceptable.